

# Fossil snakes (Squamata, Serpentes) from the tar pits of Venezuela: taxonomic, palaeoenvironmental, and palaeobiogeographical implications for the North of South America during the Cenozoic/Quaternary boundary

Silvio Onary[1], Ascanio D. Rincón[2] and Annie S. Hsiou[1]

[1] Departamento de Biologia, Laboratório de Paleontologia de Ribeirão Preto, Faculdade de Filosofia Ciências e Letras de Ribeirão Preto, Universidade de São Paulo, Ribeirão Preto, São Paulo, Brazil
[2] Centro de Ecología, Laboratório de Paleontología, Instituto Venezolano de Investigaciones Científicas, Caracas, Venezuela

## ABSTRACT

**Background**. Tar seep deposits in South America historically are well-known for their rich record of fossil mammals, contrasting with only a few formal reports of reptile remains. Here we report a new snake fauna recovered from two tar pits from Venezuela. The fossil remains come from two localities: (a) El Breal de Orocual, which comprises an inactive tar seep estimated to be Plio/Pleistocene in age; and (b) Mene de Inciarte, an active surface asphalt deposit with an absolute age dating to the late Pleistocene.
**Methods**. The taxonomic identity of all specimens was assessed via consultation of the relevant literature and comparison with extant specimens. The taxonomic assignments are supported by detailed anatomical description.
**Results**. The Mene de Inciarte snake fauna comprises vertebral remains identified as the genus *Epicrates* sp. (Boidae), indeterminate viperids, and several isolated vertebrae attributable to "Colubridae" (Colubroidea, *sensu Zaher et al., 2009*). Amongst the vertebral assemblage at El Breal de Orocual, one specimen is assigned to the genus *Corallus* sp. (Boidae), another to cf. *Micrurus* (Elapidae), and several others to "Colubrids" (Colubroides, *sensu Zaher et al., 2009*) and the Viperidae family.
**Conclusions**. These new records provide valuable insight into the diversity of snakes in the north of South America during the Neogene/Quaternary boundary. The snake fauna of El Breal de Orocual and Mene de Inciarte demonstrates the presence of Boidae, Viperidae, "colubrids", and the oldest South American record of Elapidae. The presence of *Corallus*, *Epicrates*, and viperids corroborates the mosaic palaeoenvironmental conditions of El Breal de Orocual. The presence of Colubroides within both deposits sheds light on the palaeobiogeographical pattern of caenophidians snake colonization of South America and is consistent with the hypothesis of two episodes of dispersion of Colubroides to the continent.

Corresponding author
Silvio Onary, silvioonary@usp.br

## INTRODUCTION

Tar seeps represent a unique taphonomic and preservational context for the recovery of fossils, often providing unparalleled insight into the history of past biotas (*LaDuke, 1991a*; *Friscia et al., 2008*; *Solórzano, Rincón & McDonald, 2015*; *Brown, Curd & Anthony, 2017*). These sites are generally interpreted as entrapment areas, with exemplar deposits, where mainly mammalian carnivore and associated herbivore taxa were recovered (*Brown, Curd & Anthony, 2017*). Besides the representative mammalian fauna, these peculiar deposits also often yield small vertebrates, plants, and invertebrates (e.g., insects) in a lagerstätten-type condition (*LaDuke, 1991a*; *Ward et al., 2005*; *Friscia et al., 2008*; *Rincón et al., 2009*; *Rincón, Prevosti & Parra, 2011*; *Solórzano, Rincón & McDonald, 2015*; *Holden et al., 2015*; *Holden et al., 2017*).

Venezuela contains several tar pits, however, only two have been paleontologically explored: El Breal de Orocual (*Czaplewski, Rincón & Morgan, 2005*; *Rincón, 2006*; *Rincón, White & McDonald, 2008*; *Rincón, Alberdi & Prado, 2006*; *Rincón et al., 2009*; *Rincón, Prevosti & Parra, 2011*; *Holanda & Rincón, 2012*) and Mene de Inciarte (*Rincón, White & McDonald, 2008*; *Prevosti & Rincón, 2007*; *Steadman, Oswald & Rincón, 2015*). The majority of reports detailing the palaeodiversity of these deposits have focused on large mammals, e.g., canids, proboscids, felids, and xenarthrans (*Prevosti & Rincón, 2007*; *Rincón, Alberdi & Prado, 2006*; *Rincón & White, 2007*; *Rincón et al., 2009*; *Rincón, Prevosti & Parra, 2011*; *Holanda & Rincón, 2012*; *Solórzano, Rincón & McDonald, 2015*), contrasting with relatively few reports of small vertebrates and reptiles (*Brochu & Rincón, 2004*; *Czaplewski, Rincón & Morgan, 2005*; *Fortier & Rincón, 2013*; *Steadman, Oswald & Rincón, 2015*; *Onary-Alves, Hsiou & Rincón, 2016*).

The interval recorded by these deposits covers key geological periods, representing some of the major palaeobiogeographical and palaeoenvironmental transitions within South America. The late Pliocene/ early Pleistocene (El Breal de Orocual) is chronologically linked with the establishment of the continental connection between the Central and South America continents (*Ituralde-Vinent et al., 2000*) and thus the beginning of the Great American Biotic Interchange (GABI) (*Woodburne, Cione & Tonni, 2006*). On the other hand, the late Pleistocene (Mene de Inciarte) is well-known for the dramatic climatic changes that occurred throughout the globe at this time (*Peizhen, Molnar & Downs, 2001*). The interaction between these factors shaped the palaeoenvironmental and palaeobiogeographical histories of the groups inhabiting this region (*Simpson, 1930*; *Woodburne, Cione & Tonni, 2006*). However, most treatments of this history have been strongly biased towards the mammalian fossil record (*Simpson, 1930*). In this contribution, we report on the fossil snakes from two tar pits from Venezuela, discussing their palaeobiogeographical and taxonomic implications. This partially fills a crucial gap in the Pliocene fossil snake record, increasing our understanding of squamate diversity during the Neogene/ Quaternary boundary in the North of South America.

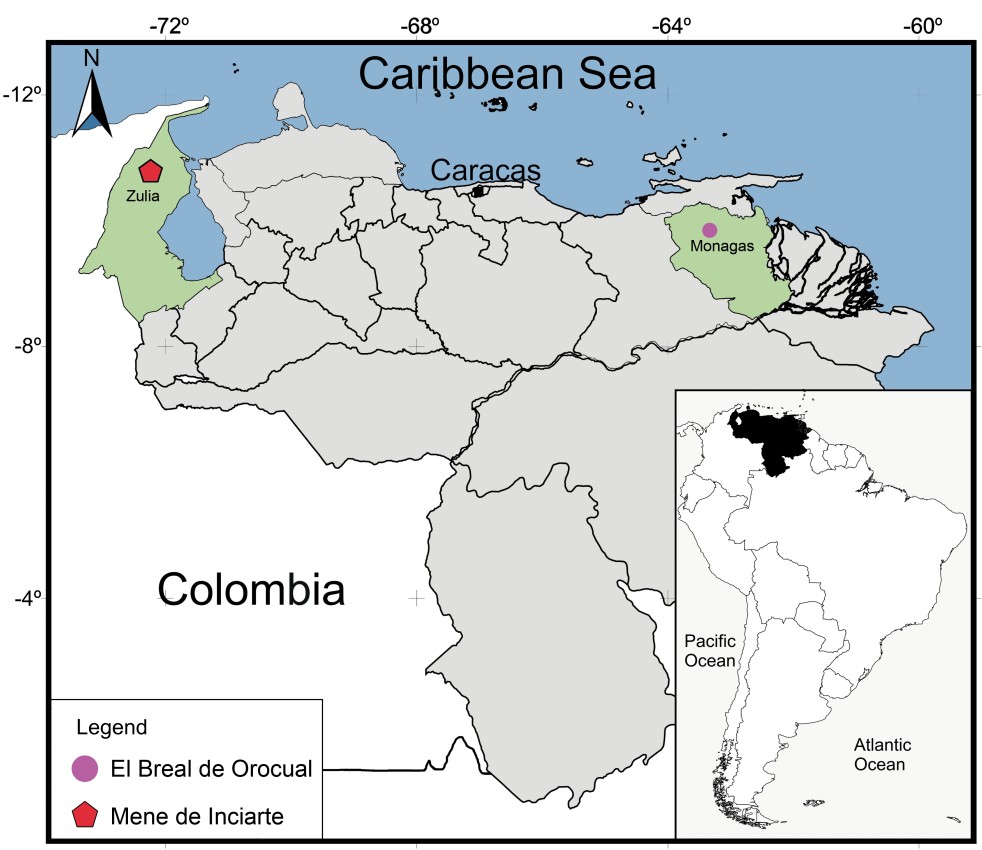

**Figure 1** **Geographical map of Venezuela showing the relative position of the deposits where the snake remains were found.** El Breal de Orocual (Plio/ Pleistocene), in pink dot, and Mene de Inciarte (upper Pleistocene) in red pentagon. (Map drawing by Ascanio Rincón and minor edits by Silvio Onary).

# GEOLOGICAL SETTINGS

## El Breal de Orocual

The recovered fossil material comes from an inactive tar seep deposit, located nearly 20 km from Maturín County, Monagas state, north eastern Venezuela (Fig. 1). The locality is situated within the Mesa Formation (*Hackley et al., 2006*; *Rincón et al., 2009*) and consists of a series of open asphalt fissures, of which one has been extensively explored (ORS16 of *Solórzano, Rincón & McDonald, 2015*; the site of this study). The tar pit has not been dated absolutely; however, the Mesa Formation was estimated by thermoluminescence (TL) to range from ∼2 Ma to 0.5 Ma (early to middle Pleistocene; *Carbón, Schubert & Vaz, 1992*). Alternatively, the 30 identified taxa from the ORS16 vertebrate fossil assemblage strongly suggests an age of late Pliocene–early Pleistocene, particularly with respect to the occurrence of *Smilodon gracilis* (Carnivora, Felidae) and cf. *Chapalmatherium* (Rodentia, Hydrochoeridae), which are considered characteristically Pliocene/Pleistocene taxa (*Rincón et al., 2009*; *Solórzano, Rincón & McDonald, 2015*). Therefore, in this contribution we follow the Plio–Pleistocene age (∼2.6 Ma) for the El Breal de Orocual deposit, based on both biostratigraphy (*Rincón et al., 2009*; *Rincón, Prevosti & Parra, 2011*;

*Holanda & Rincón, 2012*; *Solórzano, Rincón & McDonald, 2015*) and geological evidence that suggests an age of greater than 2.0 Ma for the tar pit (see dating issues discussed in *Carbón, Schubert & Vaz, 1992*).

### Mene de Inciarte

Mene de Inciarte is an active surface asphalt with production of consolidate sediments and liquid oil (*Steadman, Oswald & Rincón, 2015*). It is located in Mara County, Zulia state, northwest of Venezuela, about 90 Km from Maracaibo in the lower hills of Sierra de Perijá (Fig. 1) (*Czaplewski, Rincón & Morgan, 2005*; *Rincón, White & McDonald, 2008*; *Steadman, Oswald & Rincón, 2015*). Previous geochronological studies of the asphalt seep estimated its formation during the Quaternary with reference to the flooding of fissures with liquid asphalt (*Urbani & Galarraga, 1991*) and relative dating based on the fossil mammal record, the latter suggesting a Pleistocene age for the deposit (e.g., pampatheriids, mastodons, equids, and ground sloths) (*McDonald, Moody & Rincón, 1999*). A more recent absolute date yields an age estimate between 25,500 ± 600 14C yr BP (28,456–30,878 cal yr BP) and 27,980 ± 370 14C years BP (31,165–32,843 cal yr BP), based on collagen samples of *Glyptodon clavipes* (Mammalia, Xenarthra) (*Jull et al., 2004*).

## MATERIAL & METHODS

**Specimens**: All examined specimens consist of vertebral remains that are housed within either the El Breal de Orocual (OR–) or Mene de Inciarte (MI–) collections of the paleontological collection of Instituto Venezoelano de Investigacíones Científicas (IVIC), Caracas, Venezuela. The fossils comprise precloacal trunk vertebrae and rarer postcloacal specimens. The manner of preservation is variable between the specimens.

**Anatomical analysis**: To provide as accurate a taxonomic assignment as possible, all material was described with reference to the relevant literature as well as comparison with extant specimens outlined in Table 1. The anatomical description follows the terminology of *Auffenberg (1963)*, *Hoffstetter & Gasc (1969)*, *Rage (1984)*, *Rage (2001)*, *Lee & Scanlon (2002)*, *Hsiou & Albino (2009)*, *Albino (2011)* and *Hsiou et al. (2014)* (Fig. 2A). Quantitative data is based on *LaDuke (1991a)* and *LaDuke (1991b)* (Fig. 2B). Measurements were taken with an analogic calliper (0.02 mm) and are given in millimetres.

## RESULTS

### Systematic palaeontology

Serpentes *Linnaeus, 1758*
Alethinophidia *Nopcsa, 1923*
Macrostomata *Müller, 1831*
Boidae *Gray, 1825*
Boinae *Gray, 1825*
*Corallus Daudin, 1803*
*Corallus* sp.
Fig. 3

**Table 1  Table of the comparative specimens consulted.** Museum abbreviations are given in the institutional abbreviations section.

| Taxon | Group | Museum and specimen number |
|---|---|---|
| *Boa constrictor* imperator | Boidae | AMNH R 155261, AMNH R 155257, AMNH R 77590, AMNH R 74737, AMNH R 57472 |
| *Boa constrictor* | Boidae | AMNH R 57467, AMNH R 57476, AMNH R 131475, AMNH R 75478, AMNH R 141144, AMNH 7204, AMNH R 75267, AMNH 7118, MCN.D, 333, MCN.D 335, MCN.D 343, MCN.D 344, MCN.D 347, MCN.D 351 |
| *Corallus caninus* | Boidae | AMNH R 57788, AMNH R 73347, AMNH R 57816, AMNH R 155265, AMNH R 169154, AMNH R 155260, AMNH R 73347, AMNH R 155264, AMNH R 139338, AMNH R 155263, AMNH R 57816 |
| *Crotallus durissus* | Viperidae | AMNH 56455, AMNH 744442 |
| *Crotallus durissus* terrificus | Viperidae | AMNH 77027 |
| *Clelia clelia* | Colubroidea | AMNH 57797 |
| *Bothrops atrox* | Viperidae | AMNH 29885 |
| *Bothrops bilineatus* | Viperidae | AMNH R 140856 |
| *Corallus* cf. *C. caninus* | Boidae | AMNH R 57804 |
| *Corallus annulatus* | Boidae | AMNH R 114496 |
| *Corallus batesi* | Boidae | UFMT-R 05362 |
| *Drymarchon corais* couperi | Colubroidea | AMNH R 155299 |
| *Eunectes murinus* | Boidae | AMNH 57474, MCN.D 306, MCN.D 316, MCN.D 319, MCN.D 342 |
| *Epicrates crassus* | Boidae | MCN-PV DR 0003 |
| *Epicrates striatus* | Boidae | AMNH R 140542 |
| *Epicrates striatus* striatus | Boidae | AMNH R 155262 |
| *Epicrates striatus* strigilatus | Boidae | AMNH 155259, AMNH R 70263, AMNH R 155259 |
| *Epicrates striatus* fosteri | Boidae | AMNH R 77633, AMNH R 77057 |
| *Corallus cropanii* | Boidae | AMNH R 92997 |
| *Corallus hortulanus* cookii | Boidae | AMNH R 141098, AMNH R 74832, AMNH R 7812, AMNH R 75740, AMNH R 57809 |
| *Corallus hortulanus* | Boidae | AMNH 104528, AMNH R 57786, MCN-PV DR 0001, UFMT 02389, UFMT 02398 |
| *Chironius carinatus* | Colubroidea | AMNH 82841 |
| *Dipsas indica* | Colubroidea | AMNH 53780 |
| *Drymoluber dichrous* | Colubroidea | AMNH 55847 |
| *Dendrophidian nucale* | Colubroidea | AMNH 138461 |
| *Erythrolamprus mimus* micrurus | Colubroidea | AMNH 109828 |
| *Erythrolamprus bizona* | Colubroidea | AMNH 90018 |
| *Epicrates angulifer* | Boidae | AMNH R 77596, AMNH R 114497 |
| *Epicrates cenchria* | Boidae | AMNH R 114716, AMNH R 57473, AMNH R 71153, AMNH R 75796, AMNH R 75795, MCN-PV DR 0002 |
| *Epicrates inornatus* | Boidae | AMNH 70023 |
| *Helicops angulatus* | Colubroidea | AMNH R 139137, AMNH R 155310, AMNH R 56031 |
| *Hydrodynastes bicinctus* | Colubroidea | AMNH 60822 |

**Table 1** (*continued*)

| Taxon | Group | Museum and specimen number |
| --- | --- | --- |
| *Hydrodynastes gigas* | Colubroidea | AMNH 57956 |
| *Mastigodryas boddaerti* boddaerti | Colubroidea | AMNH R 8675 |
| *Micrurus spixi* obscurus | Elapidae | AMNH 74813 |
| *Micrurus lemniscatus* diutius | Elapidae | AMNH 78969 |
| *Pseustes poecilonotus* | Colubroidea | AMNH 85309 |
| *Ninia atrata* | Colubroidea | AMNH R 75825 |
| *Oxybelis aeneus* | Colubroidea | AMNH R 155359 |
| *Oxyrhopus petola* | Colubroidea | AMNH 77649 |
| *Oxyrhopus trigeminus* | Colubroidea | AMNH 85969 |
| *Urotheca multilineata* | Colubroidea | AMNH R 98288 |
| *Spillotes pullatus* | Colubroidea | AMNH R-155390 |
| *Xenodon rhabdocephalus* | Colubroidea | AMNH 70257 |
| *Xenodon severus* | Colubroidea | AMNH 35997, AMNH R 76573 |

**Referred material**: An isolated posterior precloacal vertebra (IVIC OR–6113).

**Locality and age**: Tar Pit ORS16, El Breal de Orocual, Monagas State, Venezuela. Age estimated to be late Pliocene–early Pleistocene based on the palaeofaunal assemblage (*Rincón et al., 2009*; *Rincón, Prevosti & Parra, 2011*; *Solórzano, Rincón & McDonald, 2015*).

**Description**: The vertebra is dorsoventrally high, mediolaterally wide and anteroposteriorly short, with its vertebral centrum smaller than the neural arch width (naw > cl). In anterior view, the zygosphene is thick and dorsoventrally inclined, being wider than the cotyle (zw > ctw). The prezygapophyseal articular facets are oriented parallel to the horizontal plane. The prezygapophyseal process is short and extends beyond the prezygapophyseal articular facet. The neural canal is subtriangular. The cotyle is circular, with similar measurements of height and width (ctw ∼ cth). The paracotylar fossae are deep and do not show evidence of paracotylar foramina. The paradiapophyses are lateroventrally oriented, showing a clear distinction between the dia–and parapophyseal articular facets.

In posterior view, the lateral edges of the neural arch are characteristically vaulted. Although the zygantrum is eroded, the probable zygantral foramen is nonetheless observable as a deep excavation within the zygantral surface. Laterally to the zygantrum, there is a series of small round pits filled with sediment, which here are interpreted as parazygantral foramina (*sensu Lee & Scanlon, 2002*). The postzygapophyses are transversely level with the horizontal plane. The condyle has a marked circular outline morphology (cnw ∼ cnh).

In lateral view, the neural spine rises from the anterior margin of the zygosphene roof. It is anteroposteriorly short, exceeding from the posterior margin of the neural arch. The zygosphene articular facets are oval and dorsolaterally oriented. Only a single lateral foramen is observable on each side of the centrum. The vertebral centrum is anteroposteriorly short with a well-marked precondylar constriction. The condyle, although distorted, is convex and slightly deflected dorsally. Ventrally, the haemal keel originates at the cotyle, extending posteroventrally until the level of the precondylar constriction.

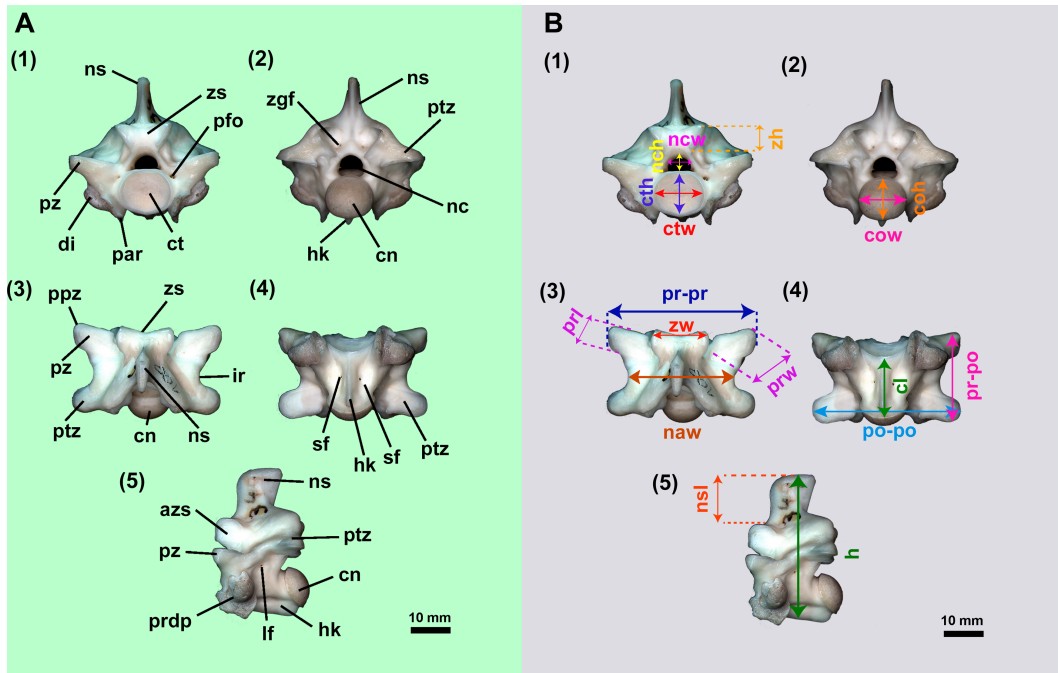

**Figure 2** **Isolated midtrunk vertebra of *Boa constrictor* evidencing the anatomical traits and quantitative data here analysed.** (A) Isolated midtrunk vertebra of *Boa constrictor* (MCN.D. 344) showing the anatomical nomenclature herein adopted. (B) same vertebra evidencing the quantitative measurements adopted in this study. Based on *LaDuke (1991a)* and *LaDuke (1991b)*. In (1) anterior, (2) posterior, (3) dorsal, (4) ventral, and (5) lateral views. Scale bar: 10 mm. Abbreviations: azs, articular facet of zygosphene; cl, centrum length; cn, condyle; coh, condyle height; cow, condyle width; ct, cotyle; cth, cotyle height; ctw, cotyle width; di, diapophysis; h, total height of vertebra; hk, haemal keel; ir, interzygapophyseal ridge; lf, lateral foramen; naw, neural arch width; nc, neural canal; nch, neural canal height; ncw, neural canal width; ns, neural spine; nsl, neural spine length; par, parapophysis; pfo, paracotylar foramen; po-po, distance between postzygapophyses; ppz, parapophyseal process; prdp, paradiapophysis; prl, prezygapophysis length; pr-po, distance between prezygapophyses and postzygapophyses of the same side; pr-pr, pr–pr, distance between prezygapophyses; prw, prezygapophysis width; ptz, postzygapophisis; pz, prezygapophysis; sf, subcentral foramen; zgf, zygantral foramen; zh, zygosphene height; zw, zygosphene width. (Photography source: Silvio Onary).

In ventral view, the centrum is anteroposteriorly short and triangular shaped. The subcentral fossae are deep and well-delimited in the anterior region of the centra. The postzygapohyses are broad and possesses subtriangular morphology.

In dorsal view, the neural arch is slightly wider than long (pr-pr > pr-po). The articular facets of the prezygapophyses are anterolaterally oriented, subtriangular in shape, and longer than wide (prl > prw). The zygosphene roof bears markedly triangular lateral lobes with a distinct slightly convex mid lobe, typifying the crenate condition (*sensu Auffenberg, 1963*). A deep interzygapophyseal ridge extends between the pre–and postzygapophysis. There is a deep posterodorsal notch in the mid portion of the posterior edge of the neural arch, which exposes a large part of the condyle.

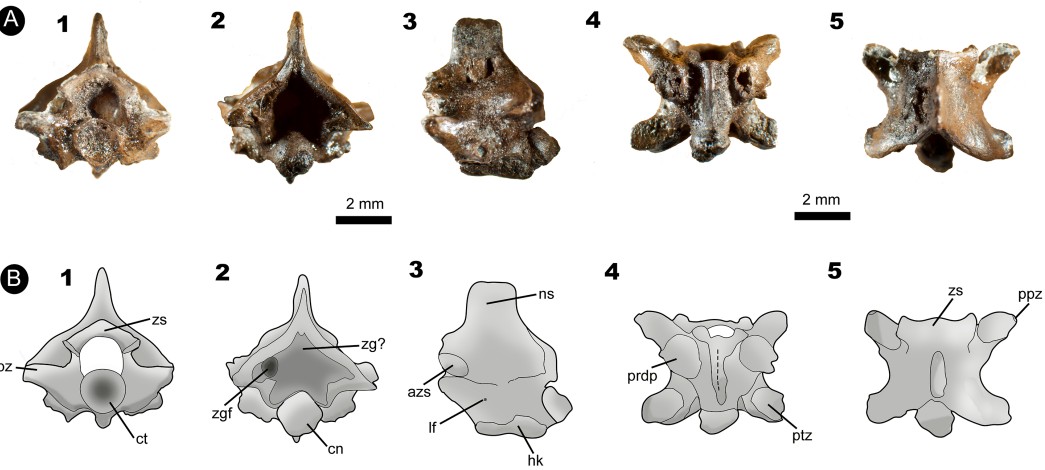

**Figure 3** **Fossil specimen IVIC OR–6113.** (A) Isolated posterior precloacal vertebra attributed to *Corallus* sp. (IVIC OR–6113). (B) Schematic drawing of the specimen evidencing its anatomical structures. In (1) anterior, (2) posterior, (3) lateral, (4) ventral, and (5) dorsal views. Abbreviations in Fig. 2. (Photography and outline drawing source: Silvio Onary).

**Measurements (in millimetres)**: IVIC OR-6113: **cl**. 3.4; **coh**.1.4; **cow**.1.3; **cth**. 0.9; **ctw**.1.0; **h**. 5.7; **naw**. 3.6; **nch**. 1.4; **ncw**. 1.2; **nsl**. 2.3; **nsh**. 2.1; **po-po**. 5.3; **pr-pr**. 5.6; **pr-po**. 4.7; **prl**. 1.6; **prw**. 1.0; **zh**. 0.9; **zw**. 2.9.

**Identification and comparison**: The specimen IVIC OR–6113 shares with Boidae the following vertebral features: dorsoventrally broad and vaulted neural arch; a well-developed and thick zygosphene; reduced prezygapophyseal process; high neural spine; well-defined precondylar constriction; inclination of the prezygapophyses less than 15°; vertebral centrum anteroposteriorly short; and presence of a haemal keel on midtrunk vertebrae (*Rage, 1984*; *Rage, 2001*; *Albino & Carlini, 2008*; *Hsiou & Albino, 2009*; *Hsiou et al., 2013*).

Among Neotropical Boinae genera, IVIC OR–6113 can be distinguished from *Eunectes* and *Boa* primarily with respect to its smaller absolute size (*Hsiou & Albino, 2010*). *Boa* also differs substantially in its more vaulted condition with a deeper posterodorsal notch (posterodorsal notch length ∼50% pr-po) (*Onary-Alves, Hsiou & Rincón, 2016*), whereas *Eunectes* displays a relatively depressed dorsoventrally neural arch (*Hsiou & Albino, 2009*).

IVIC OR–6113 can be attributed to the genus *Corallus* based on the following features: reduced absolute vertebral size (naw < 10 mm); wide, broad, and vaulted neural arch; prezygapophyses horizontally oriented (∼180°) in anterior view; crenate morphology of the zygosphene roof in dorsal view; neural spine perpendicular to the vertebral centrum; deep interzygapophyseal ridges; and the presence of small, pit-shaped parazygantral foramina (*sensu Lee & Scanlon, 2002*) (*Teixeira, 2013*).

With respect to intracolumnar variation, the specimen is consistent with the morphology of posterior midtrunk vertebrae, as supported by the reduced vertebral relative size (pr-po < 5 mm); long haemal keel; deep subcentral fossae; very short vertebral centrum; cotyle and condyle relatively circular shaped in outline; and a triangular shaped parapophyseal facet (*Teixeira, 2013*).

IVIC OR–6113 shares with posterior precloacal midtrunk vertebrae of the comparative specimens of *Corallus* (Table 1), the absolute vertebral size (pr-po < 5 mm); its anteroposteriorly elongated proportions; and the perpendicular orientation of the neural spine in relation to the vertebral centrum. In *Boa* the neural spine is oriented at a stronger dorsoventrally angle in addition to possessing both a spinal blade and laminar crest (*sensu Albino, 2011*). In contrast, *Epicrates* has high dorsoventrally neural spine (*Teixeira, 2013*). The neural spine of *Eunectes*, despite being low as in *Corallus*, it is markedly shortened anteroposteriorly (*Hsiou & Albino, 2009*).

The zygosphene of IVIC OR–6113 is similar to the midtrunk vertebrae of *Epicrates* and *Corallus*, which also exhibit a crenate morphology. In contrast, *Boa* and *Eunectes* have a dorsoventrally thicker zygosphene, in addition to the presence of a median tubercle in *Eunectes* (*Hsiou & Albino, 2009*) and a markedly concave zygosphene anterior edge in *Boa* (*Albino & Carlini, 2008*; *Onary-Alves, Hsiou & Rincón, 2016*).

Finally, IVIC OR–6113 shares exclusively with *Corallus* horizontally oriented prezygapophyseal facets, whereas in the other Neotropical boid genera these processes are slightly-to-modestly inclined relative to the horizontal plane (*Kluge, 1991*; *Rage, 2001*; *Hsiou & Albino, 2011*; *Teixeira, 2013*; *Onary-Alves, Hsiou & Rincón, 2016*).

There are eight extant species within the genus *Corallus* (*Uetz & Hošek, 2016*): *C. hortulanus* (*Linnaeus, 1758*); *C. caninus* (*Linnaeus, 1758*); *C. cookii* (*Gray, 1842*); *C. batesi* (*Gray, 1860*); *C. annulatus* (*Cope, 1875*); *C. ruschenbergerii* (*Cope, 1875*); *C. grenadensis* (*Barbour, 1914*); *C. blombergi* (*Rendahl & Vestergren, 1941*), and *C. cropanii* (*Hoge, 1953*). Among these species, three are currently found within Venezuela (*C. caninus*; *C. hortulanus*; *C. ruschenbergerii*), with only *C. ruschenbergerii* present in the area containing the fossiliferous deposit (*Rivas et al., 2012*). The lack of autapomorphic features limits a species-level identification for IVIC OR–6113. However, of the three species currently inhabiting the territory, *C. caninus* can be distinguished from IVIC OR–6113 with respect to its greater absolute dorsoventrally vertebral height (h); presence of a median tubercle on the zygosphene. In general morphology, IVIC OR–6113 shares a close similarity with *C. hortulanus* and *C. ruschenbergerii*, however, we conservatively prefer to restrict taxonomic assignment of the fossil specimen to *Corallus* sp. for the time being.

*Epicrates* Wagler, 1830
*Epicrates* sp.
Fig. 4

**Referred material**: An anterior isolated precloacal vertebra (IVIC MI–004)
**Locality and Age**: Mene de Inciarte Tar pit, Zulia state, Venezuela. Dated to $25,500 \pm 600$ [14]C years BP (28,456–30,878 cal years BP) and $27,980 \pm 370$ [14]C years BP (31,165–32,843 cal years AP), late Pleistocene (*Jull et al., 2004*).
**Description**: The vertebra is anteroposteriorly short, mediolaterally wide (naw > cl), and dorsoventrally high. In anterior view, the zygosphene dorsoventrally thick, with its articular facets laterally oriented. The width of the zygosphene exceeds the width of the cotyle (zw > ctw), with its median dorsal region present as a prominent convex border. The

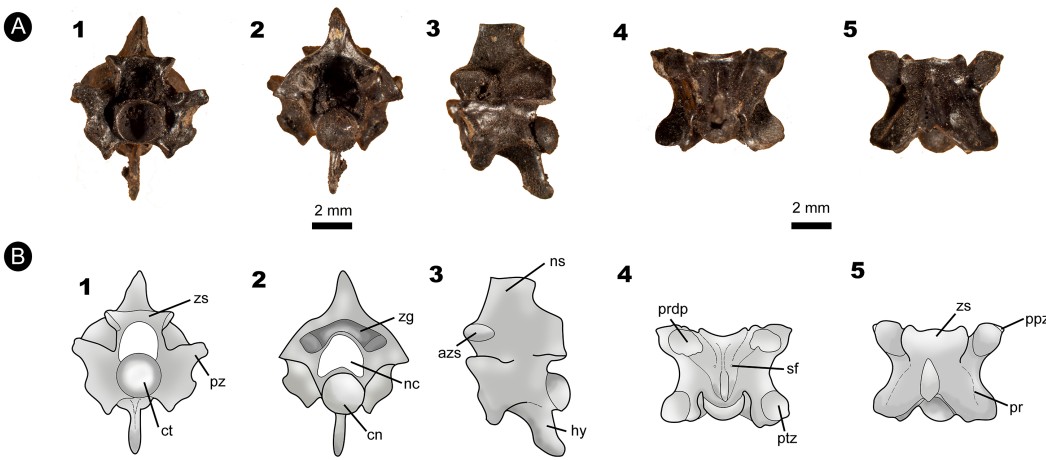

**Figure 4** **Fossil specimen of IVIC MI–004.** (A) Anterior precloacal vertebra attributed to *Epicrates* sp. (IVIC MI–004). (B) Schematic drawing of the specimen evidencing the anatomical structures. Abbreviations in the relevant section. In (1) anterior, (2) posterior, (3) lateral, (4) ventral, and (5) dorsal views. Abbreviations in Fig. 2. (Photography and outline drawing source: Silvio Onary).

prezygapophyses are oriented slightly dorsally above the horizontal axis of the centrum. A small prezygapophyseal process is located below their articular facets. The neural canal has a "trifoliate" morphology in cross-section with its width subequal to its height (ncw $\sim$ nch). The cotyle is circular shaped in outline (ctw $\sim$ cth) and displays deep paracotylar fossae but no paracotylar foramina. The paradiapophyses are broad and show a clear distinction between the dia–and parapophyseal articular facets.

In posterior view, the neural arch is strongly vaulted. The median region of the zyantrum is not preserved. The postzygapophyses of MI-004 are slightly inclined upward. The condyle is circular in shape (cow $\sim$ coh).

In lateral view, the neural spine is anteroposteriorly long, rising from the posterior edge of the zygosphene. The articular facets of the zygosphene are oval shaped and oriented dorsolaterally. The vertebral centrum of MI–004 is anteroposteriorly short and delimited by a well-marked precondylar constriction. Below the precondylar constriction there is a long hypapophysis which extends to the edge of the precondylar constriction, not exceeding beyond the posterior rim of the condyle.

In ventral view, the centrum has a marked triangular morphology tapering towards the precondylar constriction. The specimen possesses two deep subcentral fossae, with associated subcentral foramina excavating its interior on each side of the vertebra. A narrow midline keel rises from the base of the cotyle and develops posteriorly into the hypapophysis, however this process does not extend beyond the precondylar constriction. The postzygapophyses are broad (pzw > pzl) and display a subtriangular morphology.

In dorsal view, the neural arch is slightly wider than long (pr-pr > pr-po). The articular facets of the prezygapophyses are subtriangular, anterolaterally oriented, and longer than wide (prl > prw). The anterior edge of the zygosphene roof is crenate (*sensu Auffenberg, 1963*), bearing triangular lateral lobes and an anteriorly projected median lobe. Paired

parasagittal ridges (*sensu Hsiou & Albino, 2010*) extend along the roof of the neural arch from the posterior region of the zygosphene, nearly reaching the posterior margin of the neural arch. A shallow interzygapophyseal ridge extends between the pre–and postzygapophyses.

**Measurements (in millimetres)**: IVIC MI–004: **cl**:3.9; **coh**: 1.6; **cow**:2.3; **cth**:2.0; **ctw**:2.1; **h**:9.6; **naw**:4.9; **nch**:1.6; **ncw**:1.9; **nsl**:3.0; **nsh**:2.0; **po–po**:6.8; **pr–pr**:7.1; **pr–po**:5.1; **prl**:2.0; **prw**:1.4; **zh**:1.0; **zw**:3.6.

**Identification and comparison**: The specimen described above shares with the four Neotropical boid genera the following features: anterior precloacal vertebrae that are mediolaterally wide, dorsoventrally short, and dorsoventrally high relative to other aniliids and macrostomatans families; a vaulted neural arch; vertebral centrum shorter than the length of the neural arch; dorsolaterally inclination of the prezygapophysis articular facets lower than 15°; presence of a short prezygapophyseal process; deep posterodorsal notch; strong precondylar constriction; presence of paired subcentral foramina; and a mediolaterally wide and dorsoventrally thick zygosphene (*Rage, 2001*; *Lee & Scanlon, 2002*; *Szyndlar & Rage, 2003*; *Hsiou & Albino, 2009*).

IVIC MI–004 is attributed to the extant boid *Epicrates* based on the following features: small absolute size of the vertebra (h < 10 mm); vaulted neural arch; deep paracotylar fossae; dorsoventrally high neural spine; hypapophysis which does not exceed the posterior margin of the condyle; crenate zygosphene; and a centrum with a strong triangular outline in ventral (*Teixeira, 2013*).

With respect to intracolumnar variation, the fossil is interpreted as an anterior precloacal vertebra due to the presence of a well-developed hypapophysis, a feature observed exclusively in this region of the axial skeleton of boids (*Rage, 2001*); and the circular outline morphology of the cotyle and condyle (ctw ∼ cth) (*Teixeira, 2013*).

The fossil is small in absolute size (h < 10 mm), which is characteristic of the vertebrae of boids like *Corallus* and *Epicrates*, being distinct from the comparatively great vertebral size of genera as *Boa* and *Eunectes*. The vertebral height (h) of IVIC MI–004, despite the broken apex of its neural spine, is proportionally greater (i.e., considering the ratio between the neural spine size and the centrum length) than in individuals of *Corallus* and *Eunectes*. In contrast to *Boa*, the neural spine of IVIC MI–004 is relatively lower, being more similar in general size to *Epicrates*. In posterior view, IVIC MI–004 exhibits a more convexly domed neural arch compared to anterior precloacal vertebrae of *Eunectes* and *Corallus*, which exhibit a more dorsoventrally depressed morphology.

Although broken, the neural spine of IVIC MI–004 is dorsoventrally high and mediolaterally long, contrasting with *Corallus* which bears a low and mediolaterally shortened neural spine (*Hsiou & Albino, 2009*). The neural spine of *Boa,* in addition to bearing a well-delineated spinal crest and spinal blade (*sensu Albino, 2011*), exhibits a strong posterior orientation, both features that are absent in the fossil specimen.

IVIC MI–004 shares with *Corallus* and *Epicrates* the crenate morphology of the zygosphene roof (*sensu Auffenberg, 1963*); however, as pointed by *Hsiou & Albino (2010)*, this condition is variable with respect to both the individual and the position of the vertebra along the axial skeleton. Nonetheless, the crenate zygosphene of IVIC MI–004 does not

resemble the well-developed concave morphology of the zygosphene roof seen in *Boa*, nor the condition present in *Eunectes*, which possesses a median tubercle between the neural canal and the zygosphene (*Hsiou & Albino, 2009*).

Currently, two species of *Epicrates* are registered in Venezuela: *E. cenchria*, *Linnaeus (1758)* and *E. maurus*, *Gray (1849)*, of which only the distribution of *E. maurus* encompasses the Mene de Inciarte site. No autapomorphic characters of the postcranial elements have been identified as diagnostic to the specific level among the five continental species of *Epicrates* (*Riveira et al., 2011*). We therefore maintain a conservative approach and recognize IVIC MI–004 as *Epicrates* sp.

Caenophidia Hoffstetter, 1939
Endoglyptodonta *Zaher et al., 2009*
Colubroides *Zaher et al., 2009*
Colubroidea Oppel, 1811
Indeterminate genera and species
Fig. 5

**Referred material**: Four nearly complete precloacal vertebrae (IVIC OR–3667; IVIC OR–6124; IVIC OR–2618; IVIC MI–005) and one postcloacal vertebra (IVIC OR–2917).
**Localities and Age**: IVIC OR–3667; IVIC OR–6124; IVIC OR–2618: Tar Pit ORS16, El Breal de Orocual, Monagas State, Venezuela. Estimated to be late Pliocene–early Pleistocene in age based on the palaeofaunal assemblage (*Rincón et al., 2009*; *Rincón, Prevosti & Parra, 2011*; *Solórzano, Rincón & McDonald, 2015*). IVIC MI–005: Mene de Inciarte Tar pit, Zulia state, Venezuela. Dated to $25,500 \pm 600$ [14]C years BP (28,456–30,878 cal years BP) and $27,980 \pm 370$ [14]C years BP (31,165–32,843 cal years AP), late Pleistocene (*Jull et al., 2004*).
**Description**: The fossils share the following common pattern: vertebrae with the length of the vertebral centrum greater than the width of the neural arch (cl > naw). In anterior view, the neural spine is dorsoventrally high and mediolaterally thin. The zygosphene of the specimens is dorsoventrally slender, with a convex dorsal edge. The neural canal is subtriangular in shape with a tapering dorsal apex. Internally, three well-developed crests extend anteroposteriorly towards the posterior margin of the neural canal. The prezygapophyses vary in orientation among the specimens. IVIC OR–2618, IVIC OR–3667, and IVIC MI–005 show a slight dorsolaterally inclination of the prezygapophyses above the horizontal plane, whereas IVIC OR–6124 and IVIC OR–2917 exhibit a higher dorsolaterally angle of inclination, reaching the mid portion of the neural canal. The prezygapophyses are well preserved in IVIC OR–3667 and IVIC OR–6124, the main body of these processes are dorsoventrally elongate and extend well ventrally below their articular facets. The cotyles of all vertebrae are rounded with subequal width to height ratios (ctw $\sim$ cth). The paradiapophyses are anterolaterally oriented with a clear distinction between the articular facets. The pleurapophyses of IVIC OR–2917 are dorsoventrally long, mediolaterally slender, and strongly oriented ventrolaterally. The haemapophysis is positioned ventral to the cotyle and are characterized by dorsoventrally thin processes that extend a short distance along the sagittal axis of the element.

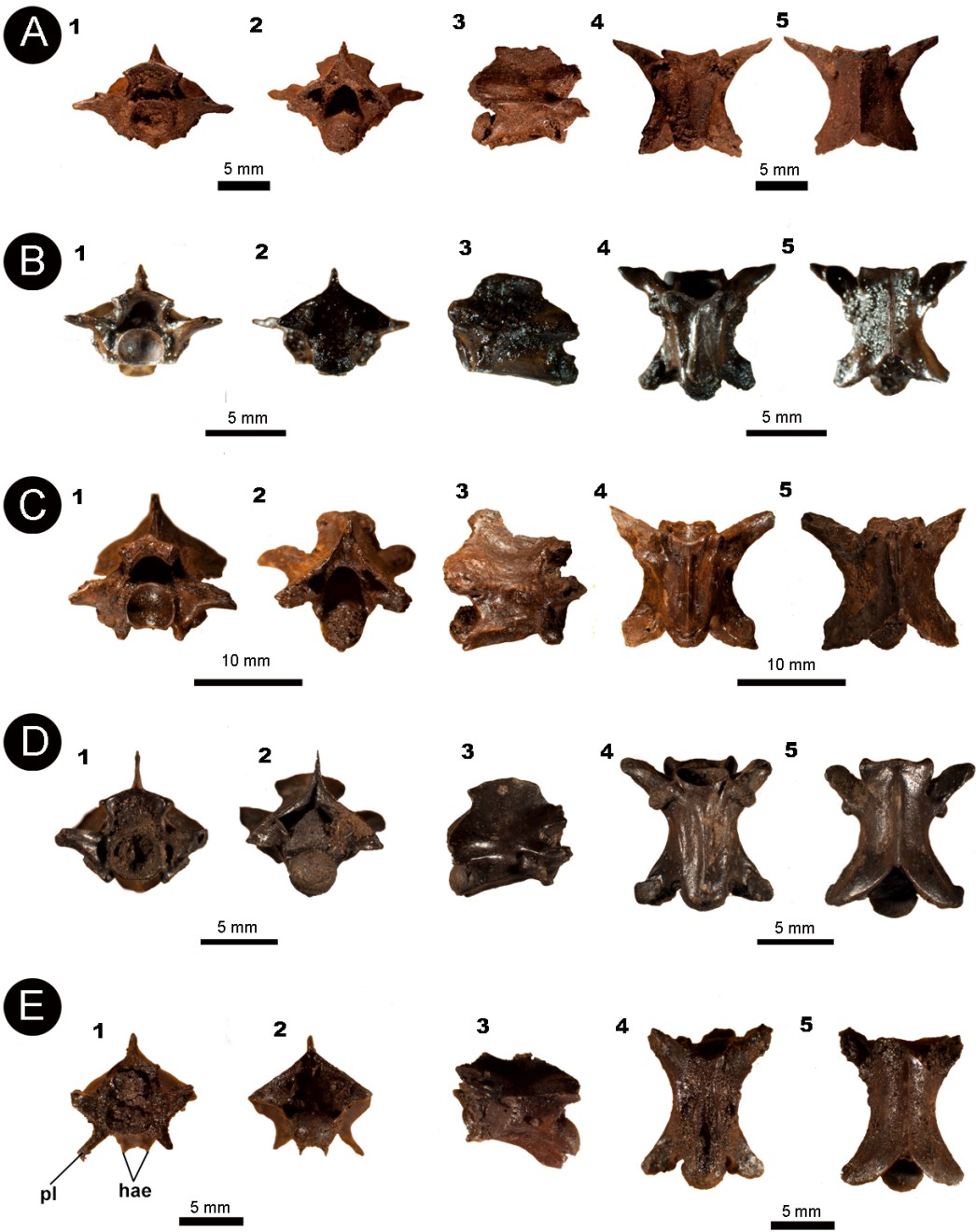

**Figure 5 Isolated vertebral remains attributed to Colubroidea.** (A) IVIC OR–3667; (B) IVIC OR–6124; (C) IVIC OR–2618; (D) IVIC MI–005; and (E) IVIC OR–2917. Abbreviations: hae, haemapophysis; pl, pleurapophysis. (Photography source: Silvio Onary).

In posterior view, the neural arches of all specimens are dorsoventrally depressed. The zygantrum mediolaterally shortened and deep with some specimens (e.g., IVIC OR–6124, IVIC OR–2618) exhibiting small paired zygantral foramina. The postzygapophyses articular facets are variable in orientation: being inclined slightly dorsolaterally in IVIC

OR–3667, IVIC OR–6124, and IVIC OR–2917; horizontally oriented in IVIC MI–005; and dorsoventrally oriented in IVIC OR–2618. The condyles of all specimens are circular in outline, with a height similar or equal to the width (cow ∼ coh).

In lateral view, the neural spine is dorsoventrally high, mediolaterally thin, and anteroposteriorly elongated. It rises from the posterior edge of the zygosphene, extending until the posterodorsal notch. Paired lateral foramina are observable on each side of the vertebral centrum only in IVIC OR–2917. The vertebral centra of all specimens are anteroposteriorly elongated. The condyle is posterodorsally inclined. With the exception of IVIC OR–2917, all specimens bear a well-developed haemal keel on ventral surface of the centrum, which does not extend beyond the condyle.

In ventral view, a prominent haemal keel and haemapophysis (in IVIC OR–6124) rise from the ventral border of the cotyle, extending longitudinally and reaching the precondylar constriction. In IVIC OR–2618 and IVIC MI–005 paired subcentral foramina can be observed on each side of the haemal keel. A marked subcentral groove excavates the mid portion of the centrum of all specimens. The postzygapophyseal articular facets are oval in shape and posterolaterally oriented in all specimens, except for IVIC MI–005 that shows a lateral orientation.

In dorsal view, the fossils are as wide mediolaterally as they are anteroventrally long (pr–pr ∼ pr–po), except for IVIC OR–2618 which is wider than long (pr–pr > pr–po). The prezygapophyseal articular facets are oval shaped (prl > prw) and anterolaterally oriented. An anterolaterally oriented prezygapophyseal process rises ventrally to the the articular facets. This process is particularly anterolaterally elongated in IVIC OR–3667. The zygosphene roof is variable among the specimens, being concave in IVIC OR–3667, straight in IVIC OR–6124, and crenate with a median lobe in IVIC MI–005 (*sensu Auffenberg, 1963*). All specimens possess a mediolaterally thin neural spine, which extends longitudinally until the posterior edge of the neural arch. The interzygapophyseal constriction is anteroposteriorly long, extending from the prezygapophyses to the articular facets of the postzygapophyses. The posterodorsal notch of the neural arch is deep in all specimens, exposing most of the cotyle.

**Measurements (in millimetres)**: *IVIC OR–3667*: **cl**:6.5; **coh**:2.6; **cow**:3.0; **cth**:2.0; **ctw**:2.5; **naw**:5.6; **nch**:2.6; **ncw**:3.0; **nsl**:5.1; **nsh**:1.9; pr–pr:9.0; **prl**:2.6; **prw**:2.1; **zh**:1.0; **zw**:4.4.
*IVIC OR–6124*: **cl**:4.9; **coh**:1.7; **cow**:2.1; **cth**:1.4; **ctw**:2.1; **h**:5.0; **naw**:3.5; **nch**:1.4; **ncw**:1.9; **nsl**:3.9; **nsh**:1.0; **po–po**:6.0; **pr–pr**:6.4; **pr-po**:6.6; **prl**:2.1; **prw**:1.1; **zh**:0.5; **zw**:3.0.
*IVIC OR–2618*: **cl**:8.0; **coh**:3.1; **cow**:3.7; **cth**:3.1; **ctw**:3.1; **naw**:7.1; **nch**:2.1; **ncw**:3.1; **po–po**:10.9; **pr–pr**:13.4; **pr–po**:11.0; **prl**:4.6; **prw**:2.4; **zh**:1.0; **zw**:5.0.
*IVIC MI–005*: **cl**:6.7; **coh**:2.5; **cow**:2.7; **cth**:2.0; **ctw**:2.2; **h**:7.1; **naw**:3.9; **nch**:2.0; **ncw**:2.2; **nsl**:5.1; **nsh**:1.9; **po–po**:7.3; **pr–po**:8.0; **prl**:2.4; **prw**:1.3; **zh**:0.7; **zw**:3.8.
*IVIC OR–2917*: **cl**:9.4; **coh**:2.8; **cow**:3.6; **cth**:3.7; **ctw**:3.9; **naw**:5.2; **po–po**:9.8; **pr–pr**:9.6; **pr–po**:11.7.

**Identification and Comments**: Colubroidea is a monophyletic group supported by several synapomorphic features that include both cranial and soft tissue characters; however, none of them relate to the axial skeleton (*Rieppel, 1988*; *Zaher, 1999*; *Zaher et al., 2009*). The group currently includes about 1,853 of the 3,596 catalogued extant snake species

(*Uetz & Hošek, 2016*), representing a well-diversified clade with a young evolutionary history (i.e., Cenozoic). The fossils described above can be attributed to Colubroidea based on the following combination of features: anteroposteriorly elongated vertebral morphology; neural arch longer than wide (cl > naw); extremely dorsoventrally slender zygosphene (zh ≤ 1 mm); dorsoventrally high neural spine; paradiapophyses with a clear distinction between their articular facets; and the presence of an anterolaterally elongated prezygapophyseal process (*Rage, 1984*; *Holman, 2000*; *Albino & Montalvo, 2006*).

Traditionally, vertebrae that display the above features have been attributed to the generic group "Colubridae". However, "Colubridae" is considered paraphyletic, with most previous analyses dealing with the group conducted using phenetic methods (*Zaher, 1999*) and therefore not representing a clade (i.e., a "natural" group) in the modern sense. For this reason, we prefer to avoid assigning anything to this generic group.

Among Colubroidea, some families are well studied, such as Calamariidae, Colubridae (clade *sensu Zaher et al., 2009*), Pseudoxenodontidae, Natricidae, and Dipsadidae (*sensu Zaher et al., 2009*). However, none of these groups have diagnoses pertaining to vertebral anatomy. It is worth noting, however, the variation in character combinations among the individual fossils, suggesting the possible occurrence of at least four different unidentified colubroidean taxa within the sample.

Endoglyptodonta *Zaher et al., 2009*
Viperidae Oppel, 1811
Indeterminate genera and species
Fig. 6

**Referred material**: One almost complete precloacal vertebra (IVIC OR–2617); three partial precloacal vertebrae (IVIC OR–6104; IVIC OR–1760; IVIC OR–3674); and a fragment of vertebral centrum (IVIC OR–5544).

**Locality and Age**: Tar Pit ORS16, El Breal de Orocual, Monagas State, Venezuela. Age estimated to be late Pliocene–early Pleistocene based on the palaeofaunal assemblage (*Rincón et al., 2009*; *Rincón, Prevosti & Parra, 2011*; *Solórzano, Rincón & McDonald, 2015*).

**Description**: In general, the vertebrae are relatively dorsoventrally high (only observable in IVIC OR–2617), slightly wider than long (pr-pr > pr-po) and have a centrum length similar to the width of the neural arch (cl ∼ naw). In anterior view, the specimens bear a dorsoventrally thin zygosphene with a straight dorsal margin. The articular facets of the zygosphene are elliptical in outline and dorsally oriented. The neural canal is trifoliate with a subequal width to length ratio (ncw ∼ nch). The articular facets of the anteroposteriorly elongate prezygapophyses are dorsolaterally inclined relative to the horizontal plane at an angle of ∼30°. The cotyles of all vertebrae are circular in outline, having a similar width to height ratio (ctw ∼ cth). Deep paracotylar fossae excavate the laterally the cotyle. The paradiapophyses, although eroded in some specimens, show a clear distinction between the articular facets. The parapophyseal processes are small, anteroventrally oriented and extend beyond the margin of the cotyle.

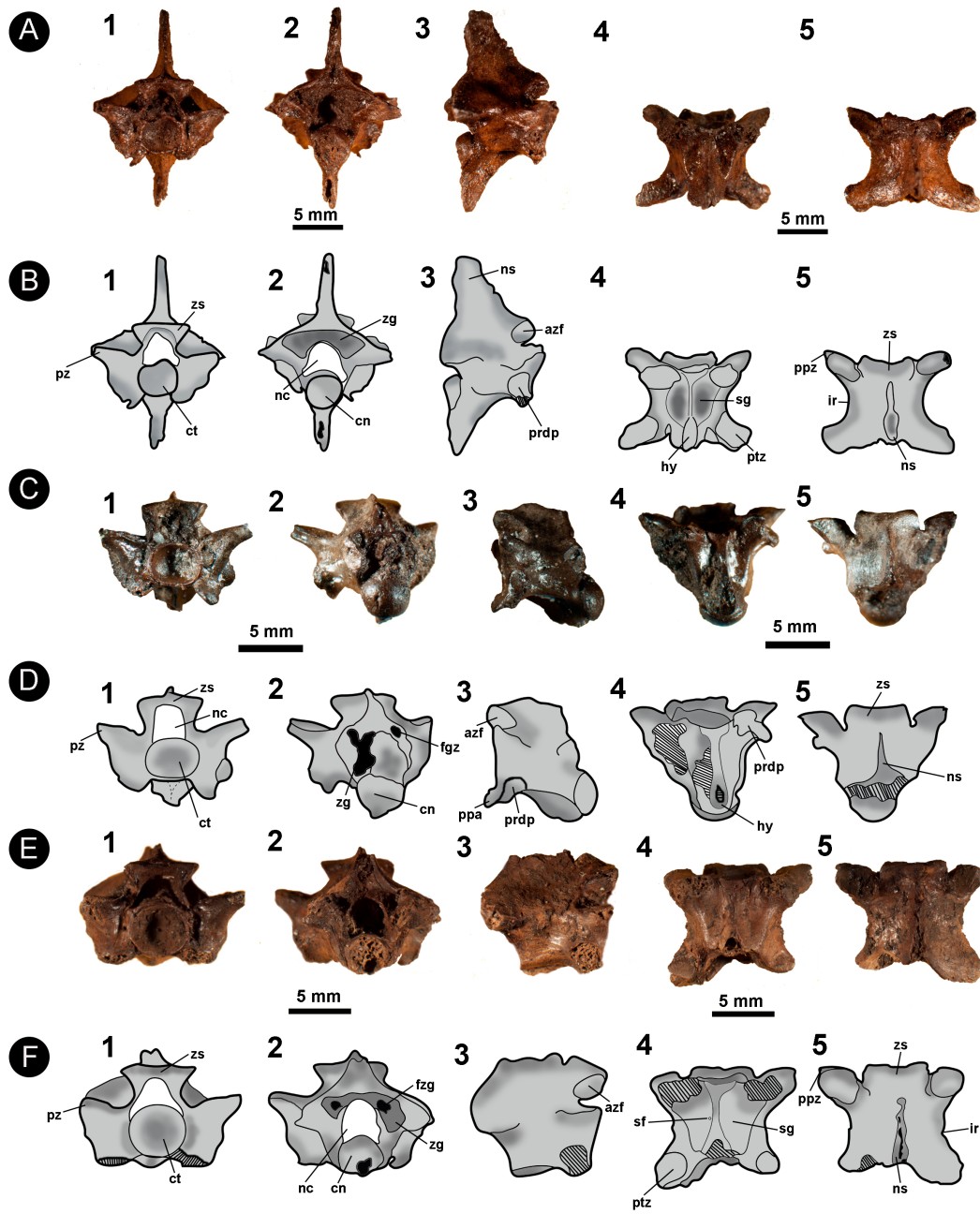

**Figure 6** **Isolated vertebral remains attributed to Viperidae.** (A) IVIC OR–2617; (B) schematic drawing of IVIC OR–2617; (C) IVIC OR–6104; (D) schematic drawing of IVIC OR–6104; (E) IVIC OR–1760; (F) schematic drawing of IVIC OR–1760. Abbreviations present in Fig. 2. (Photography and outline drawing source: Silvio Onary).

In posterior view (only preserved in IVIC OR–2617), the neural arch has a triangular outline with a shallowly concave posterior surface. The zygantrum is mediolaterally wide and deep in depth. The postzygapophyses articular facets are mediolaterally broad (pzw > pzl) and inclined slightly dorsally. In all specimens the condyle is circular in outline.

A long dorsoventrally hypapophysis rises ventrally from the condyle, exceeding its ventral margin.

Only IVIC OR–2617 preserves a neural spine. In lateral view it is well developed and dorsoventrally high. The zygosphene articular facets of all specimens are oval and dorsally oriented. The paradiapophyses are completely preserved only in IVIC OR–6104, being dorsoventrally oriented. A large parapophyseal process is observable in the anteroventral region of the parapophysis, being well developed and strongly oriented anteroventrally. The centrum bears a prominent and anteroposterioly elongated hypapophysis which in IVIC OR–2617 exceeds well beyond the posterior margin of the condyle.

In ventral view, the vertebral centrum is mediolaterally narrow and anteroposteriorly long. The subcentral fossae are variable in expression, being shallow in some specimens (e.g., IVIC–6104, IVIC OR–1760), and deep in others (e.g., IVIC OR–2616, IVIC OR–3674). In all specimens, the fossae are restricted to the anterior region of the vertebral centrum. The subcentral fossae are delimited by a well-marked subcentral margin. The hypapophysis develops longitudinally to the centrum, being broken in some specimens but clearly surpassing the posterior margin of the condyle. The articular facets of the postzygapophyses are anteroposteriorly long (pzl > pzw) and elliptical in outline.

In dorsal view, the anterior margin of the zygosphene in IVIC OR–2617 and IVIC OR–6104 is concave, whereas IVIC OR–1760 and IVIC OR–3674 exhibit a straight margin. The interzygapophyseal constriction is anteroposteriorly long and concave in shape. The neural spine of IVIC OR–2617) extends longitudinally along the dorsal surface of the arch, terminating posterior to the posterior margin of the posterodorsal notch. The prezygapophyseal articular facets are elongate ellipses (prl > prw), and oriented anterolaterally. The posterodorsal notch is deep, exposing a large portion of the condyle (only preserved in IVIC OR-2617 and IVIC OR–3674).

**Measurements (in millimetres)**: *IVIC OR-2617*. **cl**:7.0; **cth**:2.1; **ctw**:2.3; **h**:15.4; **naw**:6.0; **nch**:2.0; **ncw**:2.1; **nsl**:4.1; **nsh**:5.0; **po-po**:10.6; **pr-pr**:10.0; **pr-po**:8.0; **prl**:3.0; **prw**:1.5; **zh**:1.0; **zw**:4.8. *IVIC OR-6104*. **cl**:5.8; **cth**:2.7; **ctw**:3.0; **cth**:2.1; **ctw**:2.6; **naw**:5.5; **nch**:1.9; **ncw**:2.0; **pr-pr**:9.4; **prl**:2.3; **prw**:1.8; **zh**:0.8; **zw**:4.0. *IVIC OR-3674*. **cl**:3.2; **cth**:1.1; **ctw**:1.1; **coh**:1.9; **cow**:1.6; **naw**:6.0; **nch**:3.5; **po-po**:5.1; **pr-po**:4.9; **prl**:2.1; **prw**:1.1. *IVIC OR-3674*. **cth**:2.6; **ctw**:2.8; **naw**:6.8; **nch**:1.5; **ncw**:2.1; **prl**:2.5; **prw**:2.8; **zh**:1.4.

**Identification and Comments:** The specimens share with Colubroidea the following vertebral characters: gracile vertebrae which are longer than wide (pr–po > pr–pr); mediolatereally thin neural spine; dorsoventrally slender zygosphene; presence of prominent accessory prezygapophyseal processes; and paradiapophyses with a clear distinction between the dia–and parapophyseal articular facets (*Rage, 1984*; *Lee & Scanlon, 2002*; *Albino & Montalvo, 2006*).

The specimens possess a well-developed hypapophysis, which is considered an apomorphic character of ''Xenodermatinae'', Homalopsinae, ''Pseudoxyrhophiinae'', ''Boonodontinae'', Elapidae, Viperidae, and Natricinae (*Zaher, 1999*). Among these groups, IVIC OR–6104 shares with Viperidae a single autapomorphic postcranial character: the presence of a well-developed, strongly anteroventrally oriented parapophyseal process (*Zaher, 1999*; *Zaher et al., 2009*). Based on this character, IVIC OR–6104 is unequivocally

assigned to the Viperidae family. Despite the lack of the parapophyseal process, the other specimens can be identified as Viperidae due to the following combination of vertebral characters: a not well-elongated anteroposteriorly vertebrae (e.g., compared to Colubridae clade *sensu Zaher et al., 2009*); slender and straight zygosphene; well-developed hypapophyses; dorsoventrally depressed neural arch; postzygapophyses processes strongly oriented anterolaterally; anteroposteriorly short prezygapophyseal process; and subcentral fossae restricted to the anterior region of the centrum (*Auffenberg, 1963*; *Rage, 1984*; *Holman, 2000*; *Albino & Montalvo, 2006*; *Head, Sanchéz-Villagra & Aguilera, 2006*; *Hsiou & Albino, 2011*).

With respect to the taxonomic identity of the specimens, *Albino & Montalvo (2006)* do not recognize any diagnostic vertebral characters of Viperidae that are informative at either a the generic or specific level. Among the most common studied genera, *Camolez & Zaher (2010)* reported subtle differences between *Crotalus* and *Bothrops*, mainly regarding the morphology of the anterior margin of the zygosphene roof and the orientation of the parapophyseal processes. Among these features, the anterior margin of the zygosphene roof of *Crotalus* is generally strongly concave in its mid-region, a condition observed in IVIC OR–2617 and IVIC OR–6104.

Currently, six genera of Viperidae are distributed throughout Venezuela: *Bothrops*, *Crotalus*, *Bothriechis*, *Lachesis*, and *Porthidium*, representing 12 valid species (*Rivas et al., 2012*). Due to the lack of diagnostic vertebral features, as well the poor preservation of the specimens, here we restrict assignment of the specimens to Viperidae indet.

Endoglyptodonta *Zaher et al., 2009*
Elapoidea Boie, 1827
Elapidae Boie, 1827
cf. *Micrurus*
Fig. 7

**Referred material**: One almost complete precloacal vertebra (IVIC OR–2619).
**Locality and Age**: Tar Pit ORS16, El Breal de Orocual, Monagas State, Venezuela. Age estimated to be late Pliocene–early Pleistocene based on the palaeofaunal assemblage (*Rincón et al., 2009*; *Rincón, Prevosti & Parra, 2011*; *Solórzano, Rincón & McDonald, 2015*).
**Description**: The vertebra is relatively anteroposteriorly elongate, with a centrum length greater than the width of the neural arch (cl > naw). In anterior view, the zygosphene is convex shaped, being dorsoventrally slender and mediolaterally wider than the cotyle (zw > ctw). The neural canal is trifoliate and as wide as it is high (ncw ∼ nch). The prezygapophyses are short and oriented slightly above the horizontal plane. The only preserved prezygapophyseal process is anteroposteriorly elongated and located ventral to the right prezygapophysis. The cotyle is slightly flattened dorsoventrally such that the width is greater than the height (ctw > cth). The paradiapophyses show a clear distinction between the dia–and parapophyseal articular facets.

In posterior view the neural arch is dorsoventrally depressed. The neural spine is dorsoventrally low with its mid-region excavated by the posterodorsal notch to form a

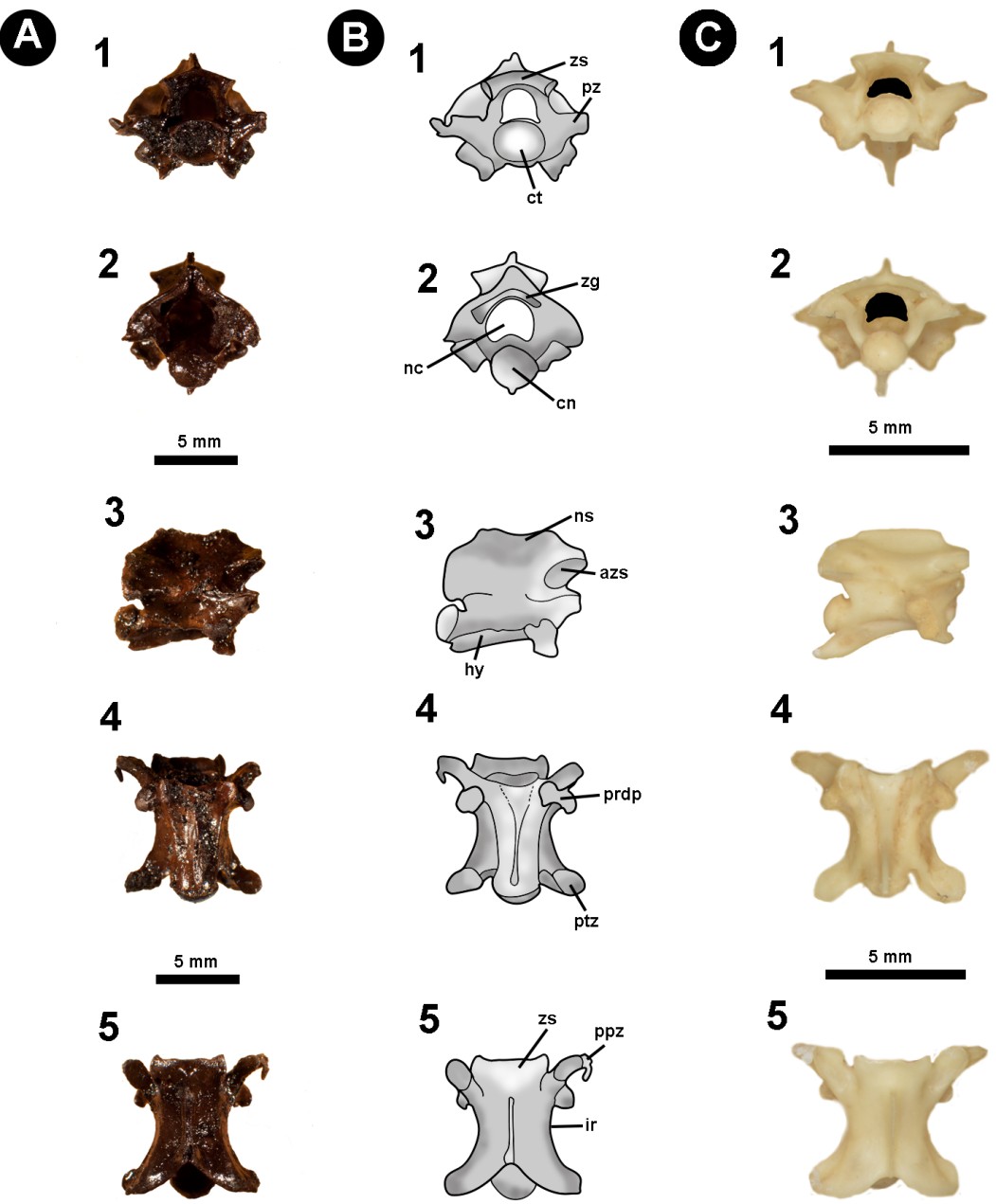

**Figure 7 Fossil specimen of IVIC OR–2619.** Isolated precloacal vertebra (IVIC OR–2619) identified as cf. *Micrurus*. (B) schematic drawing of IVIC OR–2619; (C) comparative material of precloacal vertebra of *Micrurus lemniscatus* diutius (AMNH 78969). Abbreviations in Fig. 2. (Photography and outline drawing source: Silvio Onary).

deep sulcus. The postzygapophyses are oriented slightly lateroventrally. The condyle is round with the height similar to its width (cow ∼ coh). The hypapophysis is dorsoventrally shortened and located beneath the condyle, slightly exceeding its ventral margin.

In lateral view, the neural spine is very low, dorsally straight, anteroposteriorly elongated, and sloping towards the posterior region of the neural arch. The articular facet of the

zygosphene is anterolaterally oriented and elliptical in shape. The paradiapophyses exhibit a slight anterolateral orientation. The centrum is anteroposteriorly elongated and displays a weakly developed precondylar constriction. Ventral to the centrum, the hypapophysis is mediolaterally slender with a strong posterior orientation and, despite the distal region not being preserved, probably extended beyond the posterior margin of the condyle.

In ventral view, the centrum is long and narrow (cl > naw), bearing shallow subcentral fossae which are delimited by marked subcentral margins. The hypapophysis extends longitudinally from the ventral margin of the cotyle to the mid region of the centrum, not exceeding the posterior margin of the precondylar constriction. The postzygapophyseal articular facets are elliptical in shape.

In dorsal view, the centrum has a width equal to its length (pr–pr = pr–po). The zygosphene roof possess anterolaterally tapering lateral edges and a non-crenate mid region (i.e., straight anterior edge). The prezygapophyseal articular facets are elliptical in outline (prl > prw) and orientated anterolaterally. A poorly preserved prezygapophyseal process is located ventral to the right prezygapophyseal articular facet, being mediolaterally elongate and transversely orientated relative to the prezygapophysis. Located ventral to the prezygapophyses, the diapophyseal articular facets of the paradiapophyses are convex in shape and lateroposteriorly oriented. The interzygapophyseal constriction is anteroposteriorly long, extending from the base of the prezygapophysis to the postzygapophysis, being relatively shallow. The neural spine is mediolaterally thin, rising from the posterior region of the zygosphene roof and extending longitudinally to contact the posterodorsal notch. The postzygapophyses articular facets are slightly anterolaterally oriented.

**Measurements (in millimetres)**: *IVIC OR-2619*. **cl**:5.9; **coh**:2.0; **cow**:2.2; **cth**:1.6; **ctw**:2.1; **naw**:3.6; **nch**:1.9; **ncw**:2.0; **po-po**:6.7; **pr-pr**:6.8; **pr-po**:6.8; **prl**:2.0; **prw**:0.9; **zh**:0.7; **zw**:3.7.

**Identification and Comments**: Diagnostic features for Elapidae pertain mainly to cranial characters (e.g., the morphology of the proteroglyph condition of the maxilla), in addition to morphological traits associated with the venom glands (*Underwood & Kochva, 1993*; *Zaher, 1999*). No autapomorphic postcranial features have been reported at genus/species level. Venezuela currently has two recognised genera of elapids: *Micrurus* and *Leptomicrurus* (*Rivas et al., 2012*), with two species of *Micrurus* previously reported at the fossiliferous site: *M. dissoleucus* (*Cope, 1860*) and *M. isozonus* (*Cope, 1860*).

Among the comparative osteological material accessed for this study, IVIC OR–2619 shares with the genus *Micrurus* the following vertebral characters: gracile vertebrae with a dorsoventrally depressed neural arch; oval shaped cotyle (ctw > cth); anteroposteriorly elongated pre–and postzygapophyseal articular facets (prl, pzl > prw, pzw); mediolaterally thin and very dorsoventrally low neural spine in lateral view, possessing a straight dorsal edge that develops into a slope anteriorly to the posterior margin of the neural arch; and thin hypapophysis which is strongly compressed anteroposteriorly (*Auffenberg, 1963*; *Holman, 1977*). Due to the poor preservation of the specimen, as well as the lack of formal studies concerning the postcranial osteology of Elapidae, here we prefer to restrict taxonomic attribution of IVIC OR–2619 to cf. *Micrurus*, sharing an overall vertebral morphology with

the modern genus, but lacking either diagnostic or indicative traits that can be used for more precise assignment.

## DISCUSSION

The Venezuelan snake fossil record is still scarce when compared to other South America countries (e.g., Argentina, Brazil, Colombia). With respect to Cenozoic strata, the Socorro Formation (middle Miocene) preserves *Colombophis* (Alethinophidia, *incertae sedis*), and the boid *Eunectes* (*Head, Sanchéz-Villagra & Aguilera, 2006* after *Hsiou, Albino & Ferigolo, 2010*; *Hsiou & Albino, 2010*), whereas only *Eunectes* has been reported as coming from the Urumaco Formation (middle Miocene), (*Head, Sanchéz-Villagra & Aguilera, 2006* after *Hsiou & Albino, 2010*). Recently, *Onary-Alves, Hsiou & Rincón (2016)* reported the presence of *Boa constrictor* from the El Breal de Orocual, representing the single fossil snake record for that locality. The youngest record comes from the late Pleistocene of the Cucuruchu gravels, where *Head, Sanchéz-Villagra & Aguilera (2006)* identified an indeterminate Viperidae. Although fragmentary, such occurrences provide direct insight into the palaeoenvironmental and palaeobiogeographic histories of snakes during the Cenozoic/ Quaternary in South America.

The palaeoenvironmental conditions for the North of South America have primarily been inferred with reference to the palaeofaunal mammal assemblage, which strongly suggests the predominance of dry savanna crossed by fragmentary forests, rivers, and patches of gallery forest comprised of humid–climate species of plants (*Rincón et al., 2009*; *Rincón, Prevosti & Parra, 2011*; *Solórzano, Rincón & McDonald, 2015*). The tar pit snakes corroborate the interpretation of a mosaic environmental scenario composed of small forests, arid regions, and rivers, analogous to the modern Venezuelan Llanos (*Rincón & White, 2007*; *Rincón et al., 2009*; *Rincón, Prevosti & Parra, 2011*). Although the boid genera *Corallus* and *Epicrates* are currently widespread across South America (*Henderson et al., 1995*), some species within these genera can persist only in suitable microclimatic and microenvironmental conditions, particularly forest-exclusive species (*Rodrigues, 2005*; Carvajal–Cogollo & Urbina–Cardona, 2015). Most species of *Corallus* and *Epicrates* require specific forested environments to establish a viable population (*Henderson et al., 1995*), and a major change in the microclimate can threaten these genera, even leading to local extinction (*Rodrigues, 2005*; *Carvajal-Cogollo & Urbina-Cardona, 2015*). The presence of *Corallus* in El Breal de Orocual, in addition to increasing the known boid palaeodiversity, supports the existence of forest regions with adequate environmental conditions (i.e., humidity and temperature) for habitation by boids during the Plio–Pleistocene. Moreover, the presence of Colubroides (*sensu* Zaher, 2009), such as the ''colubrids'' (Colubroidea) and especially the viperids, corroborate the existence of dry savanna components mixed with humid forested regions, since some colubrid and viperid species inhabit open areas and are well-known to live in dry environments (*e.g. Crotalus* sp.). Nowadays, *Corallus* and *Epicrates* are present in the Venezuelan Llanos (*Rivas et al., 2012*), and the record of *Corallus* during the Plio/Pleistocene, together with the presence of *Epicrates* in the Late Pleistocene, suggests that, despite climatic fluctuations, the palaeoenvironment was amenable to habitation by boids throughout this time interval.

Regarding Colubroides (*sensu Zaher et al., 2009*), an interesting biogeographical question pertains to the group's origins and entrance into South America (Figs. 8A–8C). Current palaeobiogeographical studies of the group suggest two episodes of dispersion from North America to South America, the first dating back to the uplift of the Panama Isthmus (*Albino & Montalvo, 2006*; *Hoffstetter, 1967*; *Cadle & Greene, 1993*; *Albino, 1996*), with a second episode thought to have occurred during the Plio/Pleistocene (*Wüster et al., 2002*; *Wüster et al., 2005*; *Head, Sanchéz-Villagra & Aguilera, 2006*). The oldest record of "Colubridae" in the Americas come from the late Eocene of Georgia, North America (Fig. 8A) (*Parmley & Holman, 2003*), whereas the oldest South American occurrence dates to the early Miocene of Argentina (Fig. 8B) (*Albino, 1996*). This early Miocene record, together with the late Miocene records of Viperidae from Argentina and "Colubroids" from Brazil (Fig. 8B) (*Verzi, Deschamps & Montalvo, 2004*; *Albino & Montalvo, 2006*), suggests that the first great dispersion of Colubroides occurred prior to major continental events such as the uplifting of the Panama Isthmus and the GABI (*Albino & Montalvo, 2006*; *O'Dea et al., 2016*). This dispersion can likely be explained via the aquatic crossing of a series of island complexes within Central America during the Miocene (Fig. 8B) (*Hoffstetter, 1967*; *Cadle & Greene, 1993*; *Albino, 1996*).

Based on the Venezuelan record of Viperidae in the late Pleistocene, *Head, Sanchéz-Villagra & Aguilera (2006)* suggested that Colubroides could also have reached South America during a later episode of the Neogene, mainly based on the Cucuruchu gravels record. Indeed, the combination of the Colubroides specimens described here, the fauna of Plio/Pleistocene "colubrids" and viperids at El Breal de Orocual, and the presence of a suitable colonisation route after the complete uplift of the Panama Isthmus (*O'Dea et al., 2016*), supports the hypothesis of a second entrance of Colubroides into South America at the Pliocene/Pleistocene boundary (Fig. 8C). Additionally, studies in the timing of molecular divergence (*Wüster et al., 2002*; *Wüster et al., 2005*) suggest a similar pattern in which viperids like *Bothrops*, *Lachesis*, and *Bothriechis* could have reached and diversified in South America before the total closure of the Panama Isthmus (e.g., the early Miocene records of Argentina, *Albino, 1989*; *Albino, 1996*; *Albino & Montalvo, 2006*). In contrast genera such as *Crotalus* and *Porthidium* are thought to be late dispersers, only reaching South America after the complete uplift of the Panama Isthmus (e.g., the Venezuelan Plio/Pleistocene records of "colubrids" and viperids and the late Pleistocene viperids; *Head, Mahlow & Müller, 2016* Fig. 8C). The viperid fossils of El Breal de Orocual are geographically and chronologically consistent with this later estimated entrance of *Crotalus* onto the continent (*Wüster et al., 2002*; *Wüster et al., 2005*). With respect to the described material, IVIC OR–6104 and IVIC OR–2617 bear no significant morphological distinction from extant comparative material of *Crotalus* (Table 1). These specimens share with *Crotalus* the distinct characteristic of a concave anterior edge of the zygosphene roof, which is argued to be exclusive to the genus (*Camolez & Zaher, 2010*). Despite the generic assignment of these Colubroides specimens, the material nonetheless indicates great potential for future palaeobiogeographical investigations, especially with respect to the history of viperids on the continent.

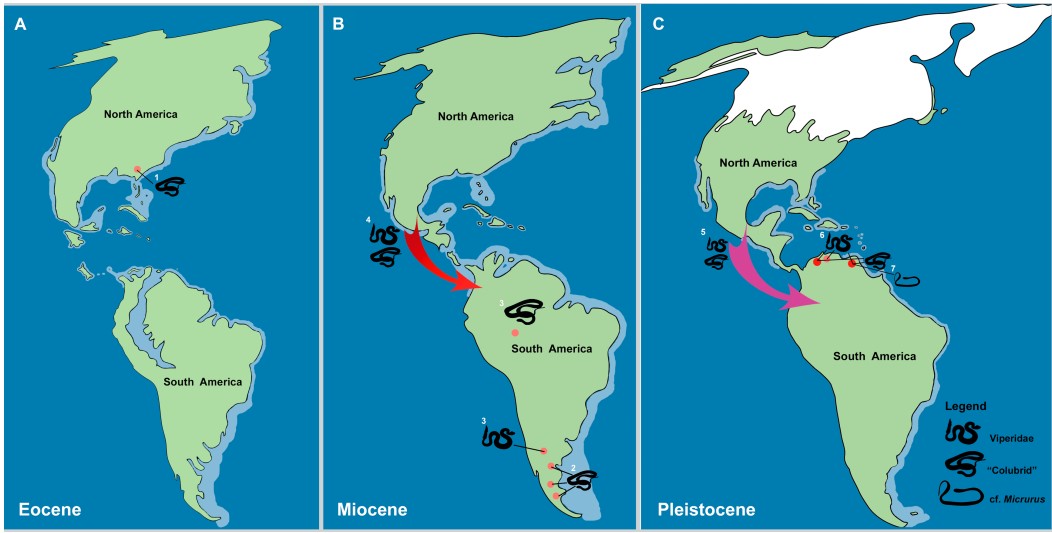

**Figure 8** **The historical biogeography of Colubroides (sensu *Zaher et al., 2009*) throughout the American continent during the Eocene to Pleistocene, based on the fossil record. (A) representative maps of Eocene; (B) Miocene; and (C) Pleistocene of America.** (1) The oldest American "Colubridae" recorded from the late Eocene of Georgia, North America (*Parmley & Holman, 2003*); (2) "Colubridae" record from the early Miocene of Argentina, South America (Colhuehuiapianense South America Land Mammal Age, SALMA) (*Albino, 1996*); (3) Viperidae remains from the late Miocene of Cerro Azul Formation (Huayquerian SALMA), Argentina (*Albino & Montalvo, 2006*) and the "Colubridae" record from the late Miocene of Amazonia, Brazil (*Hsiou & Albino, 2010*); (4) First entrance of Colubroides from North America to South America by dispersion via Central American island complex during the Miocene (*Hoffstetter, 1967*; *Cadle & Greene, 1993*; *Albino, 1996*); (5) Second event of dispersion of Colubroides from North America to South America during the Plio/Pleistocene (*Head, Sanchéz-Villagra & Aguilera, 2006*); (6) Viperidae remains from Cucuruchu gravels, late Pleistocene of Venezuela (*Head, Sanchéz-Villagra & Aguilera, 2006*); and (7) The herein described record of cf. *Micrurus* (Elapoidea, Elapidae), which is the putative oldest South American record of Elapidae, supporting the hypothesis of establishment of the genus in the South American continent at least ∼2.1 Ma, after the complete uplift of the Panama Isthmus (*Rage & Holman, 1984*). Red arrows denote the first episode of dispersion and pink arrow denote the second event of dispersion. Palaeomaps based on the reconstructions from PALEOMAP Project (*Scotese, 2010*). (Drawing designed by Silvio Onary).

The extant species of "coral snakes" are currently represented in the Americas by the genera *Micruroides* and *Micrurus* and in Asia by the genus *Sinomicrurus* (*Lee et al., 2016*). Fossil remains of "coral-snakes" are very scarce and geologically young (∼16 to 13 Ma) (*Holman, 1977*). This is concordant with the time calibrated phylogeny of the group, which estimates the divergence of the lineage at ∼30 Ma (*Lee et al., 2016*). South American records are restricted to the Quaternary of Brazil and are represented by cranial remains attributed to *Micrurus corallinus* and vertebrae assigned to *Micrurus* sp (*Camolez & Zaher, 2010*). North America preserves the oldest fossil record of the group from the late Barstovian North American Land Mammal Age of Nebraska (middle Miocene) (*Holman, 1977*), whereas material attributed to *Micrurus fulvius* and *Micrurus* cf. *M. fulvius* is known from the Pleistocene of Florida (*Auffenberg, 1963*). Records dating to the middle Miocene of Europe demonstrate the presence of the extinct *Micrurus gallicus* and *Micrurus* cf. *M. gallicus*, as well as indeterminate *Micrurus* vertebral material (*Rage &*

*Holman, 1984*; *Venczel, 2001*; *Ivanov & Böhme, 2011*). However, the palaeobiogeographical history of the genus *Micrurus* is somewhat complex and the scarcity of studies pertaining to axial skeleton anatomy hampers the identification of fossil material to a specific level, preventing further inferences about the palaeobiological past of the group (*Head, Mahlow & Müller, 2016*). Although this also impacts our knowledge of the palaeobiogeography of *Micrurus*, *Rage & Holman (1984)*, based on the fossil record, inferred a North American origin of the genus, followed by an early Miocene dispersion to Asia before eventually reaching Europe. The South American continent is estimated to have been colonized by *Micrurus* following the complete uplift of the Panama Isthmus (~2.8 Ma) (*O'Dea et al., 2016*), with dispersion potentially related to decreasing average temperatures within the higher latitudes of North America (*Rage & Holman, 1984*). The putative cf. *Micrurus* described herein is geographically and temporally consistent with the hypothesis of a South American colonization of "coral-snakes" during the Plio/Pleistocene (Fig. 8C) and represents an interesting addition to our current understanding of the biogeography of the group.

## CONCLUSIONS

The Venezuelan fossil snake record is becoming increasingly better understood, and this report contributes to our knowledge of Cenozoic squamate fossils from South America as a whole. The tar pit material described herein demonstrates the presence of several snake groups, including Boidae, Viperidae, "colubrids", and the putative oldest South American record of Elapidae. The presence of *Corallus*, *Epicrates*, and viperids, together with the previously described *Boa constrictor*, further supports the mosaic nature of the palaeoenvironment of El Breal de Orocual, being composed of forested areas together with savannah and dry open areas. The presence of Colubroides (*sensu Zaher et al., 2009*), especially the occurrence of putative fossils of *Crotalus* and cf. *Micrurus,* is consistent with the hypothesis of a second episode of dispersion and colonization of the group into South America, following the total uplift of the Panama Isthmus. This material therefore contributes genuine insight into specific palaeobiogeographic and palaeoenvironmental patterns, representing an important preliminary step. However, only identification to lower taxonomical levels can furnish more precise inferences regarding the dispersion patterns of these snake groups into South America. In this sense, the exhaustive anatomical analysis of postcranial material in addition to the application of new methodologies, such as three–dimensional morphometrics, constitutes a crucial future direction for research into this part of the palaeontological record.

### Institutional Abbreviations

| | |
|---|---|
| **AMNH** | American Museum of Natural History, New York, New York |
| **MCN.D** | Coleção Didática de Herpetologia, Museu de Ciências Naturais da Fundação Zoobotânica do Rio Grande do Sul, Porto Alegre, Brazil |
| **MCN-PV** | Seção de Paleontologia do Museu de Ciências Naturais da Fundação |
| **DR** | Zoobotânica do Rio Grande do Sul, Coleção de Paleontologia de Vertebrados, Coleção Didática de Répteis, Porto Alegre, Brazil |

| IVIC–OR | Instituto Venezoelano de Investigaciones Científicas El Breal de Orocual collection |
| IVIC–MI | Instituto Venezoelano de Investigaciones Científicas Mene de Inciarte collection; |
| UFMT | Coleção da Universidade Federal do Mato Grosso, Mato Grosso, Brazil. |

## ACKNOWLEDGEMENTS

We are deeply grateful to the following curators, staffs, and collection managers for the access and support during the development of the research: CJ Raxworthy, D Kizirian, FT Burbrink, MG Arnold, and L Vonnahme, of the Department of Herpetology (AMNH). SO thanks the following researchers of the IVIC, D Ramoni, D Rauseo, M Quiroga–Carmona, L Criollo, Y Reina, A Solórzano, M Flores, MFP Carillo, and E Catari for support while undertaking work within Venezuela. We are much indebted to HA Blain (IPHES) and J Scanlon (UNSW) for their valuable reviews, comments, language checking, and suggestions that greatly improved our manuscript. We want to acknowledge BW McPhee (FFCLRP/USP) for providing exceptional language and anatomical edits that sharply improved our work. Lastly, the authors extend special gratitude to the reviewer Jean-Claude Rage (*in memoriam*) (MNHN), whose insightful suggestions and expert observations improved not just the quality of this contribution, but the field of palaeoherpetology as a whole.

### Funding

This research was funded by Coordenação de Aperfeiçoamento de Pessoal de Nível Superior (CAPES); Richard Gilder Graduate School of American Museum of Natural History (AMNH), Collection Study Grants (grant to Silvio Onary), Conselho Nacional de Desenvolvimento Científico e Tecnológico (CNPq Proc. n. 309434/2015-7 for Annie S. Hsiou), Fundação de Amparo à Pesquisa do Estado de São Paulo (FAPESP, Process n. 2011/14080-0 for Annie S. Hsiou and Process n. 2017/00845-1 for Silvio Onary), the Venezuelan Education University, Science, and Technology Ministry (grant to Ascanio D. Rincón IVIC-1096, Venezuelan PaleoMapas); the Petróleos de Venezuela S.A. (PDVSA), and Instituto de Patrimonio Cultural, Venezuela. The funders had no role in study design, data collection and analysis, decision to publish, or preparation of the manuscript.

### Grant Disclosures

The following grant information was disclosed by the authors:
Coordenação de Aperfeiçoamento de Pessoal de Nível Superior (CAPES).
Richard Gilder Graduate School of American Museum of Natural History (AMNH).
Collection Study Grants.
Conselho Nacional de Desenvolvimento Científico e Tecnológico: 309434/2015-7.
Fundação de Amparo à Pesquisa do Estado de São Paulo: 2011/14080-0, 2017/00845-1.

Venezuelan Education University, Science, and Technology Ministry: IVIC-1096. Instituto de Patrimonio Cultural, Venezuela.

## Competing Interests

The authors declare there are no competing interests.

## Author Contributions

- Silvio Onary conceived and designed the experiments, performed the experiments, analyzed the data, contributed reagents/materials/analysis tools, prepared figures and/or tables, authored or reviewed drafts of the paper, approved the final draft, photographed, measured, and did the first hand analyses of the specimens.
- Ascanio D. Rincón contributed reagents/materials/analysis tools, authored or reviewed drafts of the paper, approved the final draft.
- Annie S. Hsiou analyzed the data, contributed reagents/materials/analysis tools, authored or reviewed drafts of the paper, approved the final draft.

## Data Availability

We declare that all recovered specimens consist in vertebral remains and are housed in the El Breal de Orocual collection (OR–) or the Mene de Inciarte collection (MI–) in the paleontological collection of Instituto Venezoelano de Investigacíones Científicas (IVIC), Caracas, Venezuela.

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
