# Peer review of "Fossil snakes (Squamata, Serpentes) from the tar pits of Venezuela: taxonomic, palaeoenvironmental, and palaeobiogeographical implications for the North of South America during the Cenozoic/Quaternary boundary"

_PeerJ, doi:10.7717/peerj.5402_

## Round 0.1 · original submission · Major Revisions

Three reviewers have checked the MS and all feel there is substance here that deserves eventual publication, but all note that the English writing needs major improvements to make it a professional and clear publication. There are also some comments about the science itself (e.g. Reviewer 1: "One argument that I completely disagree when you use the presence of Corallus and Epicrates in the Plio-Pleistocene to argue that there were “none great climatic environmental change affected the North of South America since then”")-- these must be dealt with. Please formulate a response that addresses all points of the reviewers individually in sequence, very clearly, to facilitate further review. I urge usage of an English-language writing service (or colleague(s)) to help ensure the writing is as fluid and precise as possible. We look forward to receiving the amended MS.

·

Basic reporting

I am not native English speaker, but it seems to me that this manuscript, or at least some parts, needs a language editing by a native speaker. In the present form some sentences seem to be basic translation from Spanish. There are also many errors to check in the manuscript, as line 32 “wielded” instead of “yielded”, line 191 “have an oval shape”, line 192 “On each sides of the”, line 218 “following vertebral features”, line 223 “due to its comparative reduced size”, line 233 “Regarding the intracolumnar variation”, line 304 “zygantrum” etc.

Minor comments
1. Line 695, much than localities you are speaking about countries?
2. On your figures, in my opinion, your snake vertebrae are not well positioned in posterior and anterior views. Traditional position would place the postzygapophyse articulation surfaces above the prezygapophyse ones, like it articulates in a living specimen. But it is only a small detail.
3. In the figure 8, I did not understand to what are referring the letter that are together with the snake silhouette (A to G). Maybe you can explain it in the caption?
4. Last small comment, in the whole manuscript I suggest to use mayor caps for “Tar Pit” like in the title.

Experimental design

Done

Validity of the findings

In the discussion and conclusions, there is one argument that I completely disagree when you use the presence of Corallus and Epicrates in the Plio-Pleistocene to argue that there were “none great climatic environmental change affected the North of South America since then”. It completely disagree with what is known from the global Quaternary paleoclimatic evidences. Yes, Venezuela has not had major changes as those registered in North America with important glaciations but the distribution of Epicrates and Corallus may have change owing to minor climate variations inside your study area. Please try to reformulate your argument in a not so definitive way. Because maybe great climate changes really occur in Venezuela during the Quaternary, but your snakes may have survived in small refugia and dispersed again after.

Additional comments

First of all I want to state that I am not aware of the Venezuelan snake fauna, nevertheless I read carefully the manuscript entitled “Fossil snakes (Squamata: Serpentes) from the Tar Pits of Venezuela: Taxonomical, palaeoenvironmental, and palaeobiogeographical implications”. In my opinion descriptions of the fossil vertebrae sound very well and this manuscript represents an interesting contribution to the knowledge of south American fossil snakes, even if most of attribution are made at genus level only. I strongly support its publication besides a few comments.

·

Basic reporting

The general organization of the manuscript is good, but, except in ‘Introduction’ and ‘Geological Settings’, English is poor (including many basic mistakes). I think that English is the main problem in this manuscript.
I think that the palaeoenvironmental and palaeobiogeographical contexts should be outlined in the Introduction (what was the knowledge for the concerned time interval before the present study ?). The Introduction deals only with the fossiliferous localities that yielded the studied fossils, and with similar sites, which is not the core of the study.
Descriptions are too detailed (because they are not descriptions of new taxa), parts of them are not useful.
In the Discussion, please clearly distinguish what is confirmation of previous studies and what is new. The Discussion appears to be too long and somewhat muddled. I suggest to shorten it (for example, the report of all fossil snakes from Venezuela is largely out of the scope of the paper) and to clearly state results (e.g., what are the taxonomic contents (for snakes) of the two reported waves that entered South America). The Abstract is more informative than the text itself, which is a question of drafting.
Literature appears to be as complete as possible.
Figures are relevant and of quality. A remark about captions: Anatomical abbreviations are in caption of Fig. 2, not of Fig. 3 as is stated in Fig. 3.

Experimental design

This is an original research that is within the scope of the journal. It may fill a gap (e.g. about elapids), but this should be more clearly stated.

Validity of the findings

There are novelties but they do not readily appear within the provided information (information also includes confirmations of previous works).

Additional comments

Aside from English, which should be improved, results would be more accessible to readers if parts of the text were shortened.

·

Basic reporting

There are numerous minor instances of idiomatic or grammatical errors in English, most of which do not create ambiguity and are trivial to correct; many but not all of these are marked in the annotated copy.

Background review and literature references appear fully adequate. Zaher et al. (2009) should be cited at first use of Colubroides/Colubroidea, as this is relatively new usage and may be confusing.

Structure of the ms and presentation of data is fine.

Experimental design

The ms reports original work based on previously unreported specimens, much increasing knowledge of the history of snakes in northern South America; it shows a relatively high degree of rigor in description, comparison and diagnosis.

Validity of the findings

Positively stated conclusions are justified by relevant data and arguments; the authors are appropriately cautious in discussing taxonomic assignment of specimens and implications for historical biogeography.

---

## Round 0.2 · Minor Revisions

Reviewer 3 rightly notes that the English writing still needs improvement for publication. A native/bilingual speaker's help as a coauthor or a paid editor might be the best solution. However, I am happy to report that the major scientific issues have all now been resolved.

·

Basic reporting

I have now re-review this manuscript and feel that it is much clearer than the first version. I strongly recommend it for publication, besides I saw a few erratas in the text, but I think they can be solve during proofing.

Not all Tar Pits have been capitalized in the main text. And I think that for JC Rage it must be (in memoriam) and no memorian in the Acknowledgment section.

Experimental design

ok

Validity of the findings

ok

Additional comments

no more comments

·

Basic reporting

The English grammar is still very poor in most of the text, and again I have marked (highlight, strikethrough or notes on pdf) many instances that are jarringly incorrect. I rate this as 'minor' in the context of the whole article, but not yet acceptable.

Experimental design

no comment

Validity of the findings

The faulty extrapolation noted by reviewers in the original version has been corrected, and the conclusions are now more robust.

---

## Round 0.3 · Minor Revisions

I have checked the manuscript throughout and while it has improved, the quality and clarity of the writing is still well below that which can be considered publishable, I am sorry to report.

Before we can accept the paper its entire text must be to a high standard of written English, with conjunctions, verb tenses, uses of plurals and other conventions strictly adhered to. I used the first page of the MS in the attached file to show some examples. Similar errors persist throughout the MS.

This is for the authors' own good, too, so that others can understand and cite this work- it must be clear and understandable and professional. We will have to Reject the study if is submitted again without these problems being fixed, so please attend to them carefully.

---

## Round 0.4 · accepted · Accept

Thank you for your patience and your attentive revisions. Blair McPhee is a wonderful colleague for helping you to revise this MS. I have checked the changes and am entirely satisfied with them. Well done!

#